# Role of DPP-4 and NPY Family Peptides in Gastrointestinal Symptoms Associated with Obesity and Type 2 Diabetes Mellitus

**DOI:** 10.3390/medicina61030504

**Published:** 2025-03-15

**Authors:** Mantas Malinauskas, Deimante Paskeviciene, Rūta Steponaitienė, Rita Gudaityte, Limas Kupčinskas, Anna Casselbrant, Almantas Maleckas

**Affiliations:** 1Institute of Physiology and Pharmacology, Lithuanian University of Health Sciences, 44307 Kaunas, Lithuania; 2Institute of Endocrinology, Medical Academy, Lithuanian University of Health Sciences, 50161 Kaunas, Lithuania; 3Institute for Digestive Research, Lithuanian University of Health Sciences, 50161 Kaunas, Lithuania; 4Department of Surgery, Lithuanian University of Health Sciences, 50161 Kaunas, Lithuaniaalmantas.maleckas@lsmu.lt (A.M.); 5Department of Surgery, Institute of Clinical Sciences, Sahlgrenska Academy, University of Gothenburg, 413 45 Goteborg, Sweden

**Keywords:** dipeptidyl peptidase-4, obesity, type 2 diabetes, neuropeptide Y, peptide YY, pancreatic polypeptide, gastrointestinal symptoms

## Abstract

*Background and Objectives*: Neuropeptide Y (NPY) family peptides and dipeptidyl peptidase-4 (DPP-4) are involved in gastrointestinal regulation and may contribute to obesity and type 2 diabetes mellitus (T2DM) pathophysiology. This study investigates their expression in jejunal muscular tissue and associations with gastrointestinal symptoms in patients with obesity, with (OB+/DM+) and without T2DM (OB+/DM−). *Materials and Methods*: This cross-sectional study includes forty-four patients undergoing laparoscopic Roux-en-Y gastric bypass divided based on T2DM status. Gastrointestinal symptoms were assessed using the Gastrointestinal Symptom Rating Scale (GSRS) questionnaire, and jejunal tissue samples were analyzed for DPP-4, NPY, peptide YY (PYY), and pancreatic polypeptide (PP) mRNA and protein levels. *Results*: DPP-4, NPY, PYY, and PP gene expression in jejunal muscular tissue was similar between groups. In the OB+/DM+ group, PP protein was higher, while DPP-4 and PYY were lower compared to the OB+/DM− group. Significant positive correlations between DPP-4 and NPY, PYY, and PP were found in the OB+/DM− group, while only DPP-4 and PYY correlated in the OB+/DM+ group. Gastrointestinal symptoms in the OB+/DM− group showed positive correlations with PP (abdominal pain), DPP-4 (indigestion), and NPY (constipation). *Conclusions*: The study demonstrates significant differences in DPP-4, PYY, and PP protein expression between patients with obesity, with or without T2DM. Peptide correlations with gastrointestinal symptoms in non-diabetic patients suggest distinct regulatory mechanisms, warranting further research.

## 1. Introduction

The rapid change in key elements of lifestyle such as increased stress, unhealthy eating habits and reduced physical activity contributes to the significant rise of overweight and obesity worldwide. If the current linear trend in obesity growth continues, 54% of the adult population globally will have overweight or obesity by 2035 [1]. Research has found a close association between obesity and type 2 diabetes mellitus (T2DM). Accumulation of excessive intra-abdominal, intrahepatic, intramyocellular, and pancreatic fat increases the risk of T2DM [2]. Furthermore, a population study from the US has demonstrated that weight gain is in direct relation with the risk of T2DM; for every kilogram gained the risk was increased by 7.3% [3].

T2DM is a chronic progressive disease which affects different human organs and organ systems. The most common complications of T2DM are hypertension, dyslipidemia, retinopathy, nephropathy, neuropathy, and cardiovascular disease [4]. Gastrointestinal symptoms are present in more than 70% of T2DM patients seen in outpatient clinics and may cause significant morbidity and decrease in quality of life [5,6]. It was widely accepted that autonomic neuropathy is the main underlying mechanism for gastrointestinal changes in patients with T2DM. However, with expanding knowledge, additional mechanisms were suggested such as smooth muscle myopathy, disturbances in enteric nervous system and interstitial cells of Cajal, and disruption of hormonal homeostasis [7,8]. Abnormal levels of gastrointestinal hormones such as ghrelin, cholecystokinin (CKK), glucagon-like peptide-1 (GLP-1), and peptide YY (PYY) may have significant impact on gastrointestinal secretion and motility, and can cause gastrointestinal symptoms such as nausea, diarrhea, and constipation.

Ghrelin is an orexigenic hormone produced by enteroendocrine cells (EECs) in the fundal part of the stomach and duodenum, augments appetite and food intake, and accelerates gastric emptying [9]. CKK, GLP-1, and PYY are all anorexigenic hormones; however, their impact on gastrointestinal function is slightly different. CKK secreted by I cells in the duodenum ad proximal jejunum slows gastric emptying and accelerates small intestinal transit, while GLP-1 and PYY produced by L cells in the ileum and colon are responsible for the “ileal brake” mechanism and decelerate gastric emptying [10]. PYY and pancreatic polypeptide (PP), which belong to the NPY family, are of particular interest, because they act as hormones and/or neurotransmitters, and play an important role in the gut–brain axis. Enteric neurons, specifically inhibitory motoneurons of the myenteric plexus as well as noncholinergic secretomotor neurons of the submucosal plexus, represent the major source of NPY in the intestine [11]. Within the brain, NPY is expressed in neuronal pathways spanning from brainstem to cortex [12]. PYY and PP are found exclusively within EECs in the ileum and large bowel, and could act locally or by circulating blood, or by synaptic transmission [13].

The NPY family peptides exhibit biological actions in humans on the brain and gastrointestinal tract through four Y-receptor subtypes (Y1R, Y2R, Y4R, and Y5R). In the central nervous system, NPY family peptides play important roles in regulating food intake, energy homeostasis, anxiety, mood, and depression-related behavior [12]. In small intestine and colon Y1R, Y2R, and Y4R are present in submucous and myenteric plexuses [14]. NPY and PYY have similar affinities for the Y1R, Y2R, and Y5R subtypes and exert similar biological effects on gastrointestinal tract—inhibit gastric and pancreatic secretion, delay gastric emptying, and play an important role in the ileal brake mechanism, and stimulate absorption of water and electrolytes [14]. Moreover, visceral pain seems to be mediated by peripheral Y2R and endogenous PYY has a hypoalgesic effect on visceral pain [15]. In general, PYY effects on gastrointestinal tract are more potent than NPY [16]. PP binds to Y4R and inhibits stomach and small intestine motility as well as pancreatic and intestinal water/electrolyte secretion [14]. In animals’ and humans’ small intestine, longitudinal muscle cells express only Y1R, whereas circular muscle cells express Y1R, Y2R, and Y4R [17]. NPY family peptides are metabolized by dipeptidyl peptidase 4 (DPP-4), which selectively cleaves N-terminal dipeptides. Once released, part of NPY and PYY is enzymatically cleaved by DPP-4 to NPY(3–36) and PYY(3–36) [18]. These truncated peptides lose their affinity for Y1, Y4, and Y5 receptors and become Y2R receptor agonists, whereas cleaved PP loses its biological activity through the Y4R [19] (Figure 1). Thus, DPP-4 represents an important mechanism regulating NPY family receptor specificity.

Previous studies on irritable bowel syndrome (IBS) suggest that low PYY concentration and a low density of PYY cells in the large intestine contribute to dysmotility and visceral sensitivity observed in IBS patients [13]. There is also some evidence that NPY family peptides may play an important role in the pathophysiology of inflammatory bowel disease (IBD) [14]. Plasma DPP-4 levels and activity as well as NPY levels are increased in T2DM patients as compared to healthy control individuals and are considered to play an important role in T2DM pathology [20,21]. Since DPP-4 enzymatically cleaves NPY family peptides, altering their receptor specificity, it may influence gastrointestinal function. Therefore, the main goal of the current study was to evaluate the expression of DPP-4 and NPY family peptides in the small intestine muscular tissue and explore association of these substances with gastrointestinal symptoms among patients with or without T2DM and obesity.

## 2. Material and Methods

### 2.1. Ethics Statement

The study was conducted with the approval of the Kaunas Regional Ethics Committee for Biomedical Research (Approval No. BE-2-29, issued on 1 March 2023). All subjects were informed verbally and in writing and signed an informed consent form. The study was conducted at the Department of Surgery, Kauno Klinikos Hospital, Lithuanian University of Health Sciences, Lithuania.

### 2.2. Study Participants and Sample Size

Patients undergoing laparoscopic Roux-en-Y gastric bypass for morbid obesity at the Department of Surgery, Kauno Klinikos Hospital, Lithuanian University of Health Sciences were invited to participate in the study. Inclusion criteria were BMI ≥ 35 with comorbidities or BMI ≥ 40 and age from 18 to 65 years. Exclusion criteria were prescription of DPP-4 inhibitors, type 1 diabetes or other non-type 2 forms of diabetes, severe ongoing psychiatric disorder, alcoholism and substance abuse, or end-stage organ disease. Health status was evaluated by a structured interview. All patients preoperatively underwent routine endocrinological evaluation and were in euthyroid status with normal levels of free T3 and T4. Patients were advised to lose weight preoperatively starting 2 weeks before surgery with a high-protein, but low-fat and low-carbohydrate diet. Patients were considered for the surgery and inclusion into the study only if they complied with the diet and lost not less than 5% of their total body weight.

T2DM was diagnosed as fasting plasma glucose levels which were ≥7.0 mmol/L or HbA1c ≥ 6.5%, or patients who were treated with peroral medications or insulin [22]. Hypertension was present if blood pressure was ≥140/90 mmHg or the patients were on hypertensive medications. Dyslipidemia was diagnosed when total cholesterol level was higher than 5.2 mmol/L (>200 mg/dL), triglyceride (TG) level > 2.26 mmol/L (>200 mg/dL), or high-density lipoprotein (HDL) level lower than 1.04 mmol/L (40 mg/dL) for men and <1.3 mmol/L (<50 mg/dL) for women, or the patients were using statins [23].

The goal of the present study was to explore associations between GI symptoms and DPP-4 and NPY family peptide expression in the small bowel muscular tissue. The sample size was estimated based on expected correlation coefficients R of 0.6, which determines a strong association. If the procedures were compared and conclusions were made with type I error 0.05 and power 0.8, we needed 19 patients in each group, 38 in total [24]. We included 19 patients in the OB+/DM+ group; however, due to technical reasons we were able to analyze small bowel samples only of 17 patients. Overall, 44 patients, 14 males and 30 females, with mean age 45.0 (12.5) years and BMI 44.5 (6.6) were included into the study. The patients were divided into two groups—patients with T2DM (OB+/DM+) and without T2DM (OB+/DM−).

### 2.3. GSRS

The severity of gastrointestinal symptoms among the study participants was evaluated by the self-administered Gastrointestinal Symptom Rating Scale (GSRS) questionnaire [25]. GSRS estimates the severity of the symptoms that were present over the past week. Fifteen questions are grouped to cover the severity of five syndromes such as abdominal pain (stomach discomfort or pain, hunger pains, and nausea), reflux (heartburn and acid reflux), indigestion (rumbling, bloating, belching, and flatulence), constipation (constipation, hard stools, and the feeling of incomplete evacuation), and diarrhea (diarrhea, loose stools, and urgent need for defecation). The answer to each question is graded on a 7-point Likert scale where 1 is no symptoms and 7 is the most severe symptoms. The mean values for diarrhea, indigestion, constipation, abdominal pain, and reflux were calculated.

### 2.4. Preparation of Surgical Resected Material

Full-thickness intestinal tissue was obtained from patients undergoing gastric bypass surgery for morbid obesity. A full-thickness specimen of the jejunal wall was resected between the gastro entero-entero anastomosis and the entero-entero anastomosis as the bowel loop was divided to create the Roux-en-Y construction. The mucosa/submucosa was separated from the muscularis propria by sharp dissection. Muscular tissue samples were snap frozen in liquid nitrogen and kept frozen (−80 °C) for later analysis of protein and RNA extraction procedures.

### 2.5. Protein Analysis

#### 2.5.1. Preparation of Tissue Homogenates

Fresh frozen jejunal muscle tissue samples were weighed, rinsed in cold phosphate-buffered saline (PBS), minced on ice, and lysed using Lysing Matrix D tubes (MP Biomedicals, Santa Ana, CA, USA) and cold radioimmunoprecipitation assay buffer (RIPA) lysis and extraction buffer (Thermo Scientific, Waltham, MA, USA). Tissue lysis was performed on a MagNA Lyser instrument (Roche, Mannheim, Germany) using three short bursts at 6000 rpm with a 15 s interval between each burst. Lysates were cooled on ice for 30 min with occasional vortexing and then clarified by centrifugation. The supernatant was aliquoted into separate microcentrifuge tubes for each enzyme-linked immunosorbent assay (ELISA) and immediately frozen at −80 °C.

#### 2.5.2. ELISA

DPP-4, NPY, PYY, and PP levels in tissue supernatants were quantified using ELISA kits. The ELISA plate and reagents were prepared, and the analysis was performed according to the manufacturer’s instructions: DPP-4 (E-EL-H6147, Elabscience, Shanghai, China); NPY (detects the 1–36 form, E-EL-H1893, Elabscience, China), PYY (detects the 1–36 form, RAB0413, Sigma-Aldrich, St. Louis, MO, USA); PP (E-EL-H2236, Elabscience, China). Absorbance at 450 nm was measured using a microplate reader (Multiskan GO 1.00.40 (Thermo Fisher Scientific)).

### 2.6. Gene Expression Analysis

#### 2.6.1. RNA Extraction from Muscular Tissues

Frozen tissue samples (approximately 25–30 mg) were homogenized using a MagNA Lyser instrument (Roche, Germany) prior to RNA extraction. RNA extraction was performed using an RNeasy Mini Kit (Qiagen, Hilden, Germany, Cat. No. 74104) according to the manufacturer’s instructions. RNA quality and concentration were evaluated using a NanoDrop 2000 spectrophotometer (Thermo Fisher Scientific, USA). The A260/A280 absorbance ratio of total RNA ranged from 1.9 to 2.1, indicating high purity.

#### 2.6.2. Complementary DNA Synthesis

Reverse transcription (RT) was performed using the High-Capacity cDNA Reverse Transcription Kit (Thermo Fisher Scientific Baltics, Baltics UAB, Lithuania, Cat. No. 4368814) in a 20 μL reaction volume with 0.5 μg total RNA as input, according to the manufacturer’s protocol. The amplification reaction was performed in a thermal cycler (Applied Biosystems, Foster City, CA, USA) using the following cycling profile: 25 °C for 10 min, 37 °C for 120 min, and 85°C for 5 min. After the reaction, cDNA samples were stored at −80 °C.

#### 2.6.3. Quantitative Real-Time Polymerase Chain Reaction (RT-PCR)

Quantitative RT-PCR (qRT-PCR) analysis was performed using an ABI 7500 fast real-time PCR system (Applied Biosystems, USA) to evaluate expression changes of DPP-4, NPY, PYY, and PP genes in jejunal muscular tissue. The housekeeping gene glyceraldehyde-3-phosphate dehydrogenase (GAPDH) was used as an endogenous control and a no-template sample was included as a negative control. All assays were performed in triplicate. The total volume of the qRT-PCR reaction was 10 µL:5 µL TaqMan Fast Advanced Master Mix 2X (Applied Biosystems, USA, Cat. No. 4444557), 0.5 µL TaqMan™ Gene Expression Assay 20X (Applied Biosystems, USA, Cat. No. 4331182), 1 µL cDNA template, and 3.5 µL nuclease-free water. TaqMan PCR assays were used for the detection of GAPDH, DPP-4, NPY, PYY, and PP genes: Hs99999905_m1, Hs00175210_m1, Hs00373890_m1, Hs00609210_m1, Hs00173470_m1, and Hs00358111_g1, respectively. PCR parameters were used as recommended for the rapid cycling mode: enzyme activation at 95 °C for 20 s, followed by 40 cycles of 3 s at 95 °C for denaturation and 30 s at 60 °C for annealing. Fluorescence data were converted to threshold cycle (Ct) measurements and analyzed using the comparative 2^−ΔΔCt^ method, with log_2_(2^−ΔΔCt^) values presented in the graphs to clearly depict relative changes in DPP-4, NPY, PYY, and PP mRNA expression.

### 2.7. Data Analysis Statistics

The Shapiro–Wilk test was used to assess the normality of data distribution. Consequently, the Student’s *t* test was used to evaluate the mean differences among normally distributed data (age, total cholesterol, LDL cholesterol, and triglycerides) and the Mann–Whitney U test among non-normally distributed all other variables in this study. Spearman’s rank correlation test was used to assess the relationships between DPP-4 levels in jejunal muscular tissue and NPY, PYY, and PP levels, as well as the relationships between DPP-4, NPY, PYY, and PP levels and gastrointestinal symptoms because of non-normal distribution of these variables. Statistical analyses were performed using the Statistical Package for the Social Sciences (SPSS) for Windows (version 23.0; IBM Corp, Armonk, NY, USA) and GraphPad Prism (version 9; GraphPad Software). A *p*-value of less than 0.05 was considered statistically significant.

## 3. Results

Table 1 compares the characteristics of patients with obesity and diabetes (OB+/DM+) and patients with obesity and without diabetes (OB+/DM−). Patients in the OB+/DM+ group were significantly older (52.7 ± 10.2 vs. 40.1 ± 11.5 years, *p* < 0.001), had higher fasting glucose levels (7.41 ± 2.12 vs. 5.49 ± 0.79 mmol/L, *p* = 0.011), and higher HbA1c levels (6.52 ± 1.22 vs. 5.16 ± 0.31%, *p* = 0.048). Hypertension was more prevalent in the OB+/DM+ group (17/0 vs. 17/10, *p* = 0.004), and triglyceride levels were higher (2.13 ± 0.9 vs. 1.28 ± 0.5 mmol/L, *p* = 0.049).

Table 2 compares gastrointestinal symptoms between patients with obesity and diabetes (OB+/DM+) and patients with obesity and without diabetes (OB+/DM−). The GSRS diarrhea score was higher in the OB+/DM+ group (2.04 ± 1.1) compared to the OB+/DM− group (1.58 ± 0.7), with a statistically significant difference in the proportion of patients with a diarrhea score ≥ 3 (5/11 vs. 2/25, *p* = 0.041).

### 3.1. Expression of DPP-4 Levels in Jejunal Muscular Tissue

DPP-4 protein concentration in jejunal muscular tissue was significantly higher in OB+/DM− patients (25.72 ± 6.43 ng/mL) compared to OB+/DM+ patients (21.24 ± 5.55 ng/mL; *p* < 0.05) (Figure 2b).

### 3.2. Expression of NPY Levels in Jejunal Muscular Tissue

NPY mRNA expression and protein concentration in jejunal muscular tissue showed no significant differences between OB+/DM− and OB+/DM+ patients (*p* > 0.05; Figure 3).

### 3.3. Expression of PYY Levels in Jejunal Muscular Tissue

PYY protein concentration in jejunal muscular tissue was significantly higher in OB+/DM− patients (229.18 ± 38.08 pg/mL) compared to OB+/DM+ patients (197.76 ± 38.70 pg/mL; *p* < 0.05) (Figure 4b).

### 3.4. Expression of PP Levels in Jejunal Muscular Tissue

PP protein concentration in jejunal muscular tissue was significantly lower in OB+/DM− patients (288.75 ± 185.83 pg/mL) compared to OB+/DM+ patients (455.44 ± 224.64 pg/mL; *p* < 0.05) (Figure 5b).

### 3.5. DPP-4’s Relationship with PYY, NPY, and PP in Jejunal Muscular Tissue in Patients with Obesity and Without T2DM

A positive and statistically significant correlation was observed between DPP-4 protein expression and PYY protein expression in jejunal muscular tissue (r = 0.453, *p* = 0.023; Figure 6b).

At the mRNA level, a strong positive correlation was observed between DPP-4 mRNA expression and NPY mRNA expression in jejunal muscular tissue (r = 0.755, *p* = 0.001; Figure 6d). Similarly, DPP-4 mRNA expression showed a strong positive correlation with PYY mRNA expression (r = 0.776, *p* = 0.001; Figure 6e). A moderate positive correlation was found between DPP-4 mRNA expression and PP mRNA expression in jejunal muscular tissue, which was statistically significant (r = 0.421, *p* = 0.035; Figure 6f).

### 3.6. DPP-4’s Relationship with PYY, NPY, and PP in Jejunal Muscular Tissue in Patients with Obesity and with T2DM

A strong positive correlation was found between DPP-4 mRNA expression and PYY mRNA expression, which was statistically significant (r = 0.730, *p* = 0.001; Figure 7e).

### 3.7. Association of Gastrointestinal Symptoms with mRNA and Protein Expression of DPP-4, NPY, PYY, and PP in Jejunal Muscular Tissue

In patients with obesity and without diabetes, the PP protein expression showed a positive correlation with abdominal pain (r = 0.408, *p* = 0.043; Figure 8a), while DPP-4 protein expression exhibited a positive correlation with indigestion (r = 0.407, *p* = 0.043; Figure 8b). NPY mRNA expression demonstrated a statistically significant positive correlation with constipation (r = 0.480, *p* = 0.018; Figure 8c).

## 4. Discussion

The results of the present study show that DPP-4, NPY, PYY, and PP gene expression in jejunal muscular tissue was similar between the groups. However, significant correlations between DPP-4 gene expression and NPY, PYY, and PP gene expressions were found in non-diabetic patients while only DPP-4 and PYY gene expressions correlated significantly among patients with T2DM. In diabetic patients, DPP-4 and PYY protein concentrations were significantly lower and PP protein concentration was significantly higher as compared to non-diabetic patients. NPY protein concentrations did not show significant difference between the groups. When GI symptoms were analysed, significant correlations were revealed only in non-diabetic patients between abdominal pain and PP protein concentration; indigestion and DPP-4 protein concentration; constipation and NPY gene expression in jejunal muscular tissue.

DPP-4 has been found to be expressed in a wide range of organs and tissues, including kidney, lung, placenta, intestine, heart, brain, liver, lung, skeletal system, endothelium, and various immune cells [26,27]. There is an established relationship between DPP-4 and gut microbiota in some inflammatory diseases like IBD, where higher DPP-4 activity reduces the diversity of beneficial bacteria in favor of pathological species. Such changes disrupt epithelial barrier integrity and promote inflammation further increasing DPP-4 expression [28]. Patients with T2DM may have similar pathophysiological effects because treatment with DPP-4 inhibitors in this population reduces inflammation and increases abundance of beneficial bacteria [29]. Regulation of DPP-4 expression by gut microbiota may alter also the levels of NPY, PYY, and PP. Due to the limited data on DPP-4’s role in diabetes and obesity, especially in the digestive system, this study investigated its local effects by comparing DPP-4 levels in small intestinal tissue between OB+/DM− and OB+/DM+ patients. Significantly lower DPP-4 protein concentration was found in jejunal muscular tissue of diabetic patients in this study. It is known that DPP-4 inhibitors may reduce intestinal tissue DPP-4 levels in mice [30]. Accordingly, the intake of DPP-4 inhibitors was one of exclusion criteria in our study. The experimental data suggest that DPP-4 is modulated in a tissue-specific manner and depends on hyperglycemia and insulin resistance [31]. Thus, other antidiabetic medications used in this study may have reduced DPP-4 expression indirectly through increased insulin sensitivity and lower glycemia levels. Furthermore, dietary substances have a direct impact on small intestine DPP-4 as a high-fat diet downregulates DPP-4 gene expression in mouse small intestine [31]. In the present study, all patients were on a high-protein, but low-fat diet at least 2 weeks before surgery. Thus, dietary impact on DPP-4 expression was minimized by standardized nutrition. Future studies need to explore the association of glucose and insulin levels in jejunal muscular tissue and dietary constituents with DPP-4 expression and DPP-4 activity. Notably, non-diabetic patients had significant positive correlation between indigestion and DDP-4 protein expression in jejunal muscular tissue. DPP-4 metabolizes NPY and PYY to NPY3–36 and PYY(3–36) [18]. Thus, increased DPP-4 expression may lead to higher concentrations of NPY(3–36) and PYY(3–36), which have an affinity for Y2R located on circular muscle cells, potentially contributing to the manifestation of indigestion through the suppression of critical digestive processes, such as secretion and motility, mediated by Y2R activation [26,27].

NPY plays several important roles in the smooth muscle of the small intestine, being present in both myenteric and submucosal ganglionic networks which provide extensive intrinsic innervation to the smooth muscle layers and mucosal targets [11,32]. It influences intestinal movements by modulating smooth muscle contractions, leading to reduced intestinal motility and slower food transit through the intestine [33]. There was a strong correlation between NPY mRNA and DPP-4 mRNA, as well as a moderate correlation between NPY and DPP-4 protein expression in the jejunal muscular tissue of OB+/DM−, but not among the OB+DM+. T2DM alters the balance between these two critical substances through decreased DPP-4 protein concentration in jejunal muscular tissue. A significant positive correlation between NPY gene expression and constipation was found among non-diabetic patients. Constipation as a symptom is mainly associated with changes in large intestine transit and previous studies have found that NPY immunoreactivity was increased in descending colon myenteric plexus of patients with idiopathic chronic constipation [34]. However, the results of the current study show that patients with constipation also have higher NPY gene expression in distal jejunal muscular tissue. Similarly to indigestion, constipation seems to be also mediated through Y2R activation. This finding emphasizes that constipation in non-diabetic patients is more related to NPY than PYY, and that this relation is lost in T2DM patients with decreasing expression of DPP-4. By contrast, in diabetic patients negative moderate correlation between NPY gene expression and abdominal pain was revealed. NPY inhibits the release of proinflammatory cytokines such as interleukin-1 beta (IL-1β) and nitric oxide (NO) through activation of Y1R [35]. Downregulation of NPY may cause neuroinflammation and consequently visceral hypersensitivity and abdominal pain [36].

In the present study, PYY protein levels in the diabetic group were significantly lower as compared to the non-diabetic group. Other studies have shown that serum PYY levels are lower in patients with obesity compared to healthy individuals [37]. DPP-4 metabolizes PYY(1–36) to PYY(3–36). While PYY(1–36) and PYY(3–36) have similar effects on intestinal motility [38] and secretion [11], the NPY2 receptor selectivity of PYY(3–36) results in specific functional differences. In the present study, a strong correlation was identified between PYY mRNA and DPP-4 expression in patients with obesity, both with and without type 2 diabetes. This finding suggests that T2DM does not significantly alter the interaction between PYY and DPP-4. Although the present study found slightly higher levels of PYY(1–36) protein in OB+/DM−, future studies will aim to measure PYY(3–36) levels to better evaluate the balance between PYY(1–36) and PYY(3–36) in jejunal muscular tissue. A moderate correlation between PYY protein expression and DPP-4 protein expression was observed only in the jejunal tissue of OB+/DM−. T2DM still has potential through chronic hyperglycemia, oxidative stress, and advanced glycation end products (AGEs) [39] to disrupt the physiological interaction between PYY and DPP-4; thus, further human studies should address this question. Although in the present study no significant correlation between PYY and GI symptoms was found, among diabetic patients negative moderate correlation between indigestion and PYY expression in jejunal muscular tissue was present. Previous studies have reported an association between plasma PYY(3–36) levels and the sensation of fullness [40]. Additionally, previous research has shown that low PYY concentrations and a reduced density of PYY cells in the large intestine contribute to the dysmotility and visceral sensitivity observed in IBS patients [13]. Furthermore, lower postprandial plasma PYY concentrations have been reported in patients with functional dyspepsia compared to healthy controls [41]. The findings of other authors regarding the relationship between PYY and GI symptoms are based on measurements of PYY levels in plasma, whereas in our study, PYY was analyzed in jejunal muscular tissue. This difference in the studied tissue could potentially explain the lack of correlation observed in the present study.

A statistically significant increase in PP protein levels was observed in the diabetic group in the current study. Interestingly, the levels of other peptides, including NPY and PYY, as well as their associated enzyme DPP-4, were lower in the diabetic group. PP exerts its bioactivity primarily through the Y4 receptor, regulating digestive processes by inhibiting pancreatic secretion, slowing gastric emptying and intestinal motility, modulating appetite, and maintaining energy balance [42,43,44,45,46]. PP is predominantly expressed in and released from pancreatic tissue, and according to the Human Protein Atlas, intestinal PP levels are significantly lower compared to pancreatic levels [47]. However, in this study, the protein concentration of PP in human jejunal muscular tissue was the highest among the analyzed peptides, exceeding 400 pg/mL in the OB+/DM+. PP is degraded by DPP-4; thus, the higher PP concentration might be explained by potentially reduced degradation, as DPP-4 levels in the jejunal muscular tissue were lower compared to PP levels. Previous studies revealed an independent association between PP levels and vascular complications in diabetes, and PP has been shown to affect retinal neuronal pathways critical for neuronal survival [48]. Further research should explore the potential influence of PP on myenteric plexus neurons and its role in gastrointestinal complications in patients with T2DM and obesity. The present study identified a moderate correlation between PP protein expression and abdominal pain among non-diabetic patients with obesity. PP binds to Y4R, which is only expressed on circular muscle cells of the small intestine [14] and inhibits small intestinal motility. Thus, the visceral pain in non-diabetic patients may originate due to disbalance in longitudinal and circular muscle cells activation. In diabetic patients, there was no relation between PP protein expression and abdominal pain. This could be explained by the hypothesis that T2DM downregulates Y4R in small bowel muscular tissue and increasing levels of PP do not cause contraction of small bowel circular muscles and abdominal pain of the same magnitude as in non-diabetic patients.

The current study did not investigate gut microbiota and their impact on GI symptoms. However, some emerging evidence suggests that gut microbiota composition may play one of the key roles in pathophysiology of IBS through gut motility, mucosal inflammation, increased permeability, and immune activation [49]. Bacterial metabolites may regulate secretion of NPY family peptides, GLP-1, and cholecystokinin (CCK). GLP-1 has an important role in glucose homeostasis and slows intestinal transit. Reduced serum levels of GLP-1 were found among patients with constipation caused by IBS and in animal models exanetide-4, GLP-1 agonist, reduced sensitivity to visceral pain and stress-related defecation [50,51]. The other gut hormones may have also a role in modulating GI symptoms. CCK-expressing cells were reduced in the duodenum in all IBS types and ghrelin-expressing cells were increased in IBS with diarrhea and decreased in IBS with constipation [52].

This study is among the first to investigate DPP-4 and NPY peptides in human jejunal muscular tissue among patients with or without T2DM and obesity. The correlations presented gives some insight into the pathophysiology of GI symptoms and its relation-ship to gut peptides. However, some limitations of the current study must be emphasized. First, the small sample size did not allow us to apply multiple comparison corrections and may lead to overestimation of the significance of some correlations. Second, we were unable to collect small intestinal tissue from healthy individuals operated on for cholecystectomy or hernia, because of the risk of complications while obtaining a full-thickness small intestinal wall biopsy. Such an analysis would have provided valuable insights into the changes of NPY family peptides and DPP-4 content within the jejunal muscular tissue of patients without obesity and T2DM. Third, we were unable to control for the effect of other anti-diabetic medications, except DPP-4 inhibitors, on DPP-4 or NPY family peptides expression in the small bowel muscular tissues, because we could not stop T2DM treatment in our population. Furthermore, anti-diabetic medications can cause GI side effects. However, in our study there were no significant differences in GI symptom scores between patients who were taking these medications (OB+/DM+ group) and those who have not (OB+/DM− group) suggesting that the GI side effects of antidiabetic medications, even if present, were minimal. Fourth, we did not investigate the effect of other gastrointestinal hormones such as ghrelin, CKK, and GLP-1 and gut microbiota composition on gastrointestinal symptoms. However, after a literature search, we decided to concentrate on the NPY family because of some evidence that these peptides might be related to IBS. Finally, this is a cross-sectional study; thus, we were unable to establish causality between gastrointestinal symptoms and NPY family peptides. However, the correlations we found suggest that DPP-4 and NPY family peptides in small intestine muscular tissue are related to GI symptoms in patients with obesity or obesity and T2DM. Future research should include longitudinal studies, with larger cohorts of patients, and investigate influence of gut microbiota on gut hormones and GI symptoms.

## 5. Conclusions

This study demonstrates significant differences in DPP-4 and NPY family peptide expression profiles between OB+/DM− and OB+DM+, highlighting distinct regulatory mechanisms. The findings suggest that the interplay between DPP-4 and peptides such as PYY, NPY, and PP may play a critical role in the development of obesity- and diabetes-related gastrointestinal complications. However, the cross-sectional design of this study does not allow us to establish causality between gastrointestinal symptoms and NPY family peptides; thus, further research, particularly prospective studies involving healthy controls, larger cohorts, and analyses of DPP-4 mediated peptide fragments, is needed to better understand these mechanisms.

## Figures and Tables

**Figure 1 medicina-61-00504-f001:**
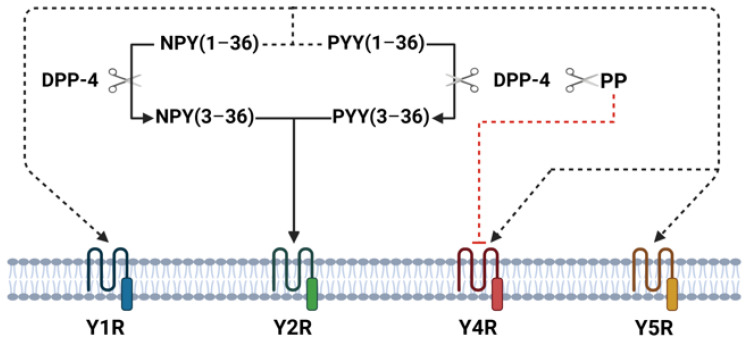
Overview of DPP-4-mediated cleavage and receptor interactions in intestinal tissue. This figure illustrates the enzymatic activity of DPP-4 in cleaving NPY(1–36), PYY(1–36), and PP, resulting in truncated forms such as NPY(3–36) and PYY(3–36). These truncated peptides shift receptor affinity, losing their binding to Y1R, Y4R, and Y5R and instead becoming agonists of Y2R. PP cleavage by DPP-4 diminishes its affinity for Y4R.

**Figure 2 medicina-61-00504-f002:**
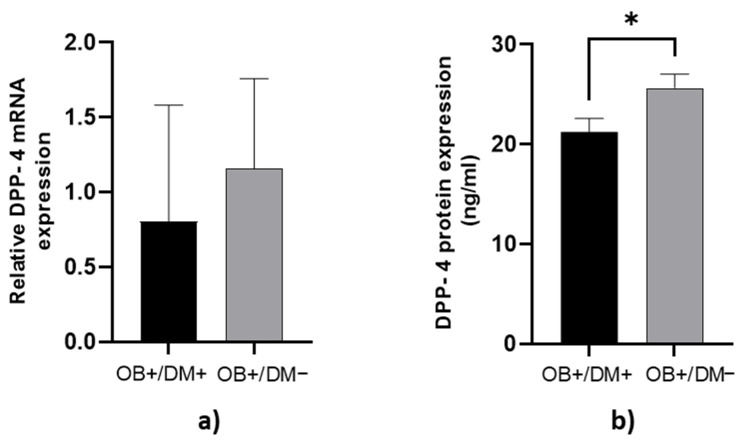
Expression of DPP-4 levels in jejunal muscular tissue. The relative DPP-4 mRNA expression in jejunal muscular tissue for patients with obesity and diabetes (OB+/DM+, *n* = 17) and patients with obesity and without diabetes (OB+/DM−, *n* = 25) (**a**). DPP-4 protein expression in jejunal muscular tissue for patients with obesity and diabetes (OB+/DM+, *n* = 17) and patients with obesity and without diabetes (OB+/DM−, *n* = 24) (**b**). Values are given as mean ± SEM. Significant differences are denoted by * *p* ≤ 0.05 (Mann–Whitney U test).

**Figure 3 medicina-61-00504-f003:**
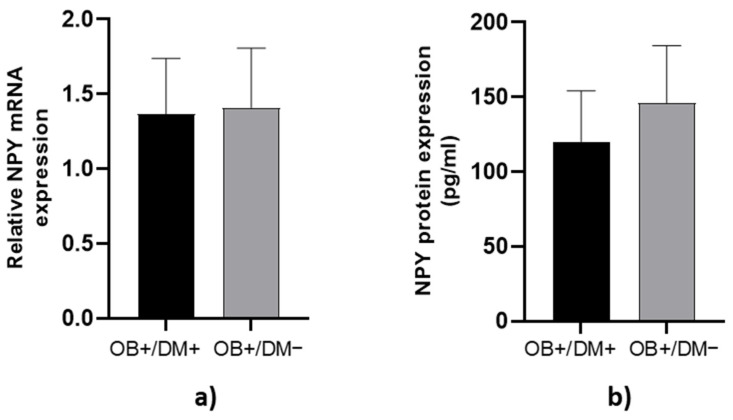
Expression of NPY levels in jejunal muscular tissue. The relative NPY mRNA expression in tissue for patients with obesity and diabetes (OB+/DM+, *n* = 24) and patients with obesity and without diabetes (OB+/DM−, *n* = 17) (**a**). NPY protein expression in tissue for patients with obesity and diabetes (OB+/DM+, *n* = 11) and patients with obesity and without diabetes (OB+/DM−, *n* = 16) (**b**).

**Figure 4 medicina-61-00504-f004:**
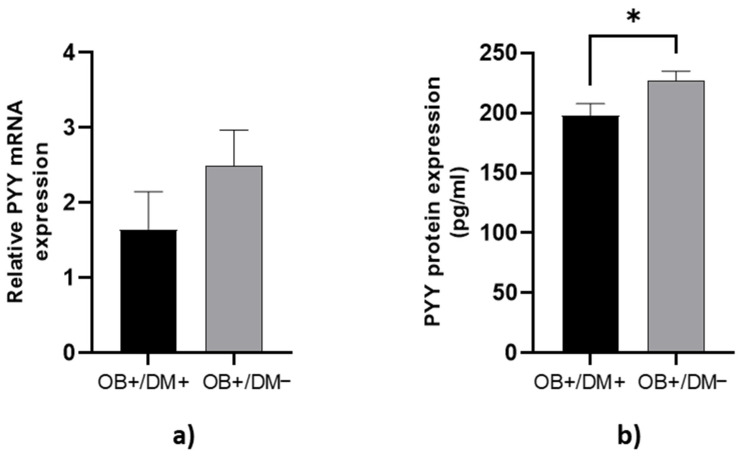
Expression of PYY levels in jejunal muscular tissue. The relative PYY mRNA expression in tissue for patients with obesity and diabetes (OB+/DM+, *n* = 17) and patients with obesity and without diabetes (OB+/DM−, *n* = 26) (**a**). PYY protein expression in tissue for patients with obesity and diabetes (OB+/DM+, *n* = 15) and patients with obesity and without diabetes (OB+/DM−, *n* = 23) (**b**). Values are given as mean ± SEM. Significant differences are denoted by * *p* ≤ 0.05 (Mann–Whitney U test).

**Figure 5 medicina-61-00504-f005:**
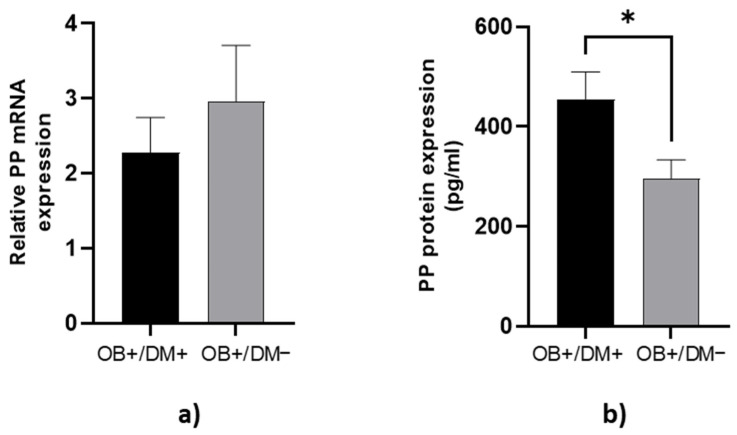
Expression of PP levels in jejunal tissue. The relative PP mRNA expression in tissue for patients with obesity and diabetes (OB+/DM+, *n* = 17) and patients with obesity and without diabetes (OB+/DM−, *n* = 26) (**a**). PP protein expression in tissue for patients with obesity and diabetes (OB+/DM+, *n* = 17) and patients with obesity and without diabetes (OB+/DM−, *n* = 25) (**b**). Values are given as mean ± SEM. Significant differences are denoted by * *p* ≤ 0.05 (Mann–Whitney U test).

**Figure 6 medicina-61-00504-f006:**
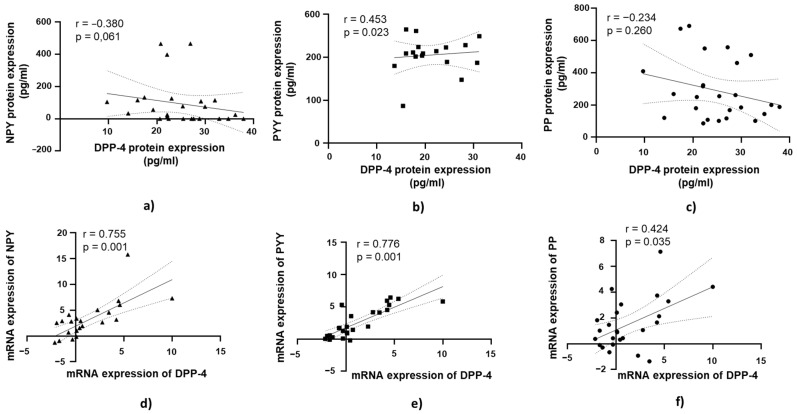
Correlation of DPP-4 protein and mRNA expression with NPY, PYY, and PP in jejunal muscular tissue in patients with obesity and without diabetes (OB+/DM−). Panels (**a**–**c**) show the correlations between DPP-4 (*n* = 24) protein expression and NPY (*n* = 16) (**a**), PYY (*n* = 23) (**b**), and PP (*n* = 25) (**c**) protein expression. Panels (**d**–**f**) illustrate the correlations between DPP-4 (*n* = 25) mRNA expression and NPY (*n* = 17) (**d**), PYY (*n* = 26) (**e**), and PP (*n* = 26) (**f**) mRNA expression. mRNA values were expressed as log2(2^−ΔΔCt^). Spearman’s rank correlation test was applied, and results are reported as correlation coefficients (r) and *p*-values (*p*).

**Figure 7 medicina-61-00504-f007:**
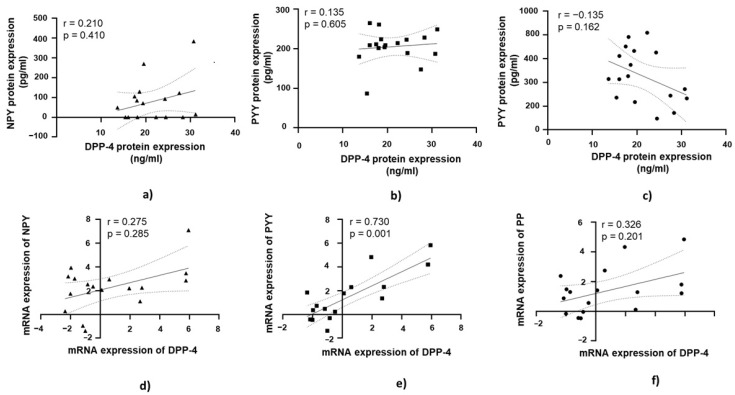
Correlation of DPP-4 protein and mRNA expression with NPY, PYY, and PP in jejunal muscular tissue in patients with obesity and diabetes (OB+/DM+). Panels (**a**–**c**) show the correlations between DPP-4 (*n* = 17) protein expression and NPY (*n* = 11) (**a**), PYY (*n* = 15) Panel (**b**), and PP (*n* = 17) (**c**) protein expression. Panels (**d**–**f**) illustrate the correlations between DPP-4 (*n* = 17) mRNA expression and NPY (*n* = 24) (**d**), PYY (*n* = 17) (**e**), and PP (*n* = 17) (**f**) mRNA expression. mRNA values were expressed as log2(2^−ΔΔCt^). Spearman’s rank correlation test was applied, and results are reported as correlation coefficients (r) and *p*-values (*p*).

**Figure 8 medicina-61-00504-f008:**
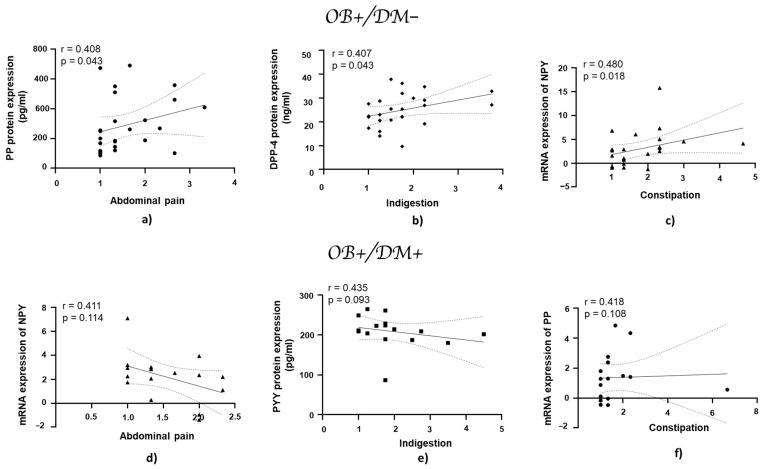
Correlation of mRNA and protein expression of NPY, PYY, PP, and DPP-4 with gastrointestinal symptoms in jejunal muscular tissue in patients with obesity and diabetes, and with obesity and without diabetes. Panels (**a**–**c**) show correlations in patients with obesity and without diabetes (OB+/DM−): PP (*n* = 25) protein expression with abdominal pain (**a**), DPP-4 (*n* = 24) protein expression with indigestion (**b**), and NPY (*n* = 17) mRNA expression with constipation (**c**). Panels (**d**–**f**) represent correlations in patients with obesity and with diabetes (OB+/DM+): NPY (*n* = 24) mRNA expression with abdominal pain (**d**), PYY (*n* = 15) protein expression with indigestion (**e**), and PP (*n* = 17) mRNA expression with constipation (**f**). mRNA values used for correlations were expressed as log2(2^−ΔΔCt^). Spearman’s rank correlation test was applied to evaluate relationships, and the results are reported as correlation coefficients (r) and *p*-values (*p*).

**Table 1 medicina-61-00504-t001:** Patients’ characteristics.

	OB+/DM+ (*n* = 17)	OB+/DM− (*n* = 27)	^1^ *p* Value
Sex M/F	7/10	7/20	0.290
Age, years ^2^ (SD)	52.7 (10.2)	40.1 (11.5)	<0.001 ***
BMI kg/m^2^ (SD)	46.3 (5.9)	43.3 (6.9)	0.141
Fasting glucose	7.41 (2.12)	5.49 (0.79)	0.011 **
HbA1c	6.52 (1.22)	5.16 (0.31)	0.048 *
Diabetes duration	5.15 (6.2)	-	
Medications for diabetes *n* (%):			
Biguanide (Metformin)	13 (76.5)		
Secretagogue	3 (17.6)		
SGLT2 inhibitors	1 (5.9)		
GLP-1 agonist	2 (17.8)		
Insulin	1 (5.9)	-	
Hypertension ^3^ y/n	17/0	17/10	0.004 **
Dyslipidemia y/n	9/8	7/20	0.070
Total cholesterol	4.61 (1.0)	5.09 (0.9)	0.312
HDL cholesterol	1.29 (0.6)	1.30 (0.2)	0.967
LDL cholesterol	2.70 (0.8)	3.23 (0.9)	0.178
Triglycerides	2.13 (0.9)	1.28 (0.5)	0.049 *

^1^ Significant difference are denoted by (*** *p* ≤ 0.001) (** *p* ≤ 0.01) (* *p* ≤ 0.05) (Mann–Whitney U test); ^2^ Standard deviation (SD); ^3^ y/n (yes/no)—presence or absence of indicated condition.

**Table 2 medicina-61-00504-t002:** Gastrointestinal symptoms among the groups.

	OB+/DM+ (*n* = 16)	OB+/DM− (*n* = 27)	^1^ *p* Value
GSRS Pain score	1.54 (0.5)	1.66 (0.9)	0.612
GSRS Pain score ≥ 3, ^2^ y/n	0/16	2/25	0.265
GSRS Reflux score	1.66 (0.6)	1.76 (1.0)	0.705
GSRS Reflux score ≥ 3, y/n	0/16	5/22	0.067
GSRS Indigestion score	1.94 (1.0)	1.75 (0.7)	0.475
GSRS Indigestion score ≥ 3, y/n	2/14	2/25	0.578
GSRS Diarrhea score	2.04 (1.1)	1.58 (0.7)	0.091
GSRS Diarrhea score ≥ 3, y/n	5/11	2/25	0.041 *
GSRS Constipation score	1.73 (1.4)	1.79 (0.9)	0.860
GSRS Constipation score ≥ 3, y/n	1/15	3/24	0.596

^1^ Significant difference are denoted by (* *p* ≤ 0.05) (Mann–Whitney U test); ^2^ y/n (yes/no)—presence or absence of indicated condition.

## Data Availability

The original contributions presented in this study are included in the article. Further inquiries can be directed to the corresponding author.

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
