# Peer review of "Role of DPP-4 and NPY Family Peptides in Gastrointestinal Symptoms Associated with Obesity and Type 2 Diabetes Mellitus"

_medicina, 2025, doi:10.3390/medicina61030504_

Round 1
Reviewer 1 Report
Comments and Suggestions for Authors
This study investigates the expression of dipeptidyl peptidase-4 (DPP-4) and neuropeptide Y (NPY) family peptides in jejunal muscle tissue of obese patients with and without Type 2 Diabetes Mellitus (T2DM) and their association with gastrointestinal (GI) symptoms. The study is important in the field of endocrinology, metabolism, and gastroenterology as it investigates the mechanisms that may contribute to obesity-associated GI symptoms and metabolic derangement.
The article is well-structured and scientifically sound, but I believe several methodological limitations and presentation issues need to be addressed before it can be considered for publication. My comments regarding the current manuscript are as follows:
Major Concerns
1. The study is cross-sectional, which limits the ability to establish causality between DPP-4, NPY peptides, and GI symptoms. The authors should acknowledge this limitation more clearly in the discussion.
2. The sample size (n = 44) is relatively small, making it prone to type II errors and limiting generalizability. Were any power calculations performed to determine the adequacy of the sample size?
3. The study excluded patients taking DPP-4 inhibitors, but other antidiabetic medications (e.g., SGLT2 inhibitors, GLP-1 receptor agonists) may also affect gastrointestinal peptides. Almost all patients in the OB+/DM group were taking metformin or one of the GLP-1 analogs. These agents are known to cause gastrointestinal side effects. How do the authors intend to eliminate the influence of these confounders?
4. Were any dietary factors taken into account? High-fat diets may alter DPP-4 and intestinal peptide secretion and potentially affect the study results.
5. Mann-Whitney U test was used for group comparisons, but Shapiro-Wilk test for normality was only briefly mentioned.
Were parametric tests considered for normally distributed variables?
Spearman correlation was used to assess associations between DPP-4, NPY peptides, and GI symptoms, but no adjustment was made for multiple comparisons (e.g. Bonferroni correction).
What confounders were controlled for in the logistic regression analysis? (Age?, etc…)
Clarify statistical methods (adjust for multiple comparisons, identify confounders in the regression analysis).
6. Results are presented both in tables and fully rewritten in the text, making the text difficult to read. It is recommended to summarize key findings in the text without repeating all numerical data already shown in the tables. Instead, highlight only significant findings with p-values to improve readability.
7. While the study identified differences in DPP-4 and peptide levels, the biological mechanisms were not adequately investigated. The authors discuss:
How insulin resistance, chronic inflammation, and gut microbiota changes may modulate DPP-4 and NPY peptide expression.
Why are only certain peptides associated with GI symptoms in non-diabetic patients but not in the diabetic group?
The gut-brain axis mechanisms, which are crucial for understanding NPY and gastrointestinal regulation, are not discussed.
8. Thyroid function significantly influences gut motility and peptide expression.
Did the study account for thyroid function markers (TSH, free T3, free T4)?
Minor Concerns
1. Provide context between DM and DPP-4 and NPY Family Peptides in the introduction.
2. The abstract should indicate that the study is cross-sectional.
3. The conclusion should briefly mention limitations and point to future directions (e.g., prospective studies, larger cohorts).
4. Some abbreviations are presented without prior definition. Ensure that all abbreviations (e.g., GSRS, PP, Y receptors) are clearly defined upon first use.
5. There are minor grammatical errors and awkward sentence structures in certain sections. A language revision is recommended to improve fluency.
6. Clearly highlight the limitations and strengths of the study in the final section of the discussion.
Strengths:
Novel investigation of DPP-4 and NPY peptides in obesity and T2DM
Use of jejunal muscle tissue samples for direct analysis
Correlations between gut peptides and GI symptoms provide insights into pathophysiology
Limitations:
Small sample size (n=44) may lead to underestimated findings.
Cross-sectional design limits causality.
Multiple comparison corrections were not applied in statistical analysis
Dietary factors, thyroid function, and gut microbiota were not taken into account
In conclusion, Based on the identified methodological concerns, statistical issues, and lack of mechanistic discussion, I recommend a major revision before the manuscript is considered for publication. If authors address all of these issues, the manuscript will improve significantly in scientific rigor, clarity, and impact.
Comments on the Quality of English LanguageAs I mentioned above.
Author Response
Please see the attachment:
Responses to reviewer comments - "Reviewer 1"

Reviewer 2 Report
Comments and Suggestions for Authors
This study investigated the expression of dipeptidyl peptidase-4 (DPP-4) and neuropeptide Y (NPY) family peptides (PYY, PP) in jejunal muscular tissue and their correlation with gastrointestinal symptoms in obese patients with and without type 2 diabetes mellitus (T2DM). This research is timely, as gastrointestinal (GI) dysfunction in T2DM is a poorly understood but clinically relevant condition. However, some areas require further improvement. The sample size was relatively small and the study did not include an independent validation cohort. Furthermore, although correlations were described, causation cannot be assumed because of the cross-sectional nature of the study. The methods are well described but can be improved with greater clarity on confounding factors, such as medication use, dietary intake, and gut microbiome composition. Despite these limitations, the results contribute to the existing literature and can be used as a basis for future mechanistic studies. See below for comments.
•There is no comparison with other GI hormones (e.g. as GLP-1 and ghrelin), which restricts the wider physiological context.
1. Introduction
• The research hypothesis is stated clearly at the end of the introduction.
•Some references are slightly outdated (e.g., obesity projections should include 2023-2024 data).
•Comparisons with other GI regulatory peptides (e.g., GLP-1 and ghrelin ) are lacking.
2. Methods
Areas for Improvement:
•Sample size justification is poorly documented. Therefore, a power analysis should be conducted.
•Potential confounders (e.g., diet, gut microbiota, and physical activity) were not addressed.
•No reference to RNA integrity checks: Was RNA degradation assessed before qPCR?
3. Results
•Figure legends are not sufficiently detailed (e.g., sample sizes for protein and mRNA analyses should be clearly stated in each figure).
•Some p-values are borderline significant (e.g., DPP-4 protein expression), and caution should be exercised in the interpretation.
•No validation in an independent dataset.
4. Discussion
Areas for Improvement:
•\tThe discussion overestimates the importance of findings without reference to potential confounders.
• Lack of discussion regarding the impact of the gut microbiota on DPP-4 and NPY family peptides.
• A Comparison with GLP-1, ghrelin, and other gut hormones is lacking.
• Longitudinal studies and mechanism validation should be included in future research.
Main Flaws:
1. No independent validation cohort, comparison with an external cohort, must be included.
2 Too little discussion of confounders – diet, gut microbiota, and medication use must be considered.
3. No mechanism validation: Correlations are presented but not functional validation (e.g., studies in cells/tissues).
Minor Flaws:
1. Not all figures include information about the samples.
2. Verification of RNA integrity was not discussed.
3. Other GI hormones (e.g. as GLP-1 and ghrelin) have not been compared.
Reviewer 3 Report
Comments and Suggestions for Authors
The manuscript presents interesting and original research. I recommend including a brief discussion of negative or unexpected results, also addressing potential limitations and future research directions to provide a balanced interpretation of the study’s impact. It may help to describe how the chosen strain reflects human disease pathophysiology.
1. This study explores how dipeptidyl peptidase-4 and neuropeptide Y family peptides are expressed in jejunal muscular tissue and how they relate to gastrointestinal symptoms in obese patients with and without Type 2 Diabetes Mellitus. The goal is to determine whether these peptides behave differently in the two groups and whether their expression links to specific gastrointestinal symptoms.
2. The study tackles an important topic in endocrinology and gastroenterology by investigating how gut peptides contribute to gastrointestinal dysfunction in obesity and diabetes. It fills a gap by looking at local expression in the jejunum rather than relying solely on systemic plasma measurements. This tissue-specific focus adds a fresh perspective to metabolic and gastrointestinal interactions.
3. This study provides:
Differences in DPP-4, peptide PYY, and pancreatic polypeptide expression in the jejunum of diabetic vs. non-diabetic obese patients.
The potential mechanisms by which gut peptides contribute to metabolic and digestive dysfunctions in obesity and diabetes.
4. The study is well-structured, but a minor detail can be improved:
Include a Healthy Control Group: Without a non-obese, non-diabetic control, it’s hard to know what "normal" levels of these peptides should be.
Measure DPP-4 Activity: Understanding its enzymatic function would provide deeper insights beyond just expression levels.
Analyze NPY(3-36) and PYY(3-36): Since DPP-4 modifies these peptides, measuring the truncated forms would give a clearer picture of functional changes.
5. The conclusions align well with the presented data. The study effectively highlights differences in peptide expression and links them to gastrointestinal symptoms. However, some mechanistic explanations, such as how DPP-4 might contribute to indigestion via Y2R activation, would benefit from additional functional validation, like receptor expression analysis or in vitro assays.
6. The references are relevant.
7. The tables and figures are ok.
8. Additional Thoughts
The study notes that diabetes medications might affect DPP-4 expression but doesn’t analyze differences by medication type. A post hoc analysis of metformin or GLP-1 receptor agonist users would be valuable.
Since bariatric surgery alters gut peptide signaling, a discussion on how these findings might translate to post-surgical patients would be insightful.
Future research could investigate whether these changes in peptide expression persist after weight loss or metabolic improvements.
Round 2
Reviewer 1 Report
Comments and Suggestions for Authors
The authors have addressed all comments except for advanced statistical analyses and group comparisons, which were excluded due to the small sample size.
Reviewer 2 Report
Comments and Suggestions for Authors
No further comments!